

# Biochemical and structural controls on the decomposition dynamics of boreal upland forest moss tissues

Michael Philben[1,6], Sara Butler[1,7], Sharon Billings[2], Ronald Benner[3,4], Kate Edwards[5], and Susan Ziegler[1]

[1]Department of Earth Sciences, Memorial University, St. John's, NL, Canada
[2]Department of Ecology and Evolutionary Biology, Kansas Biological Survey, University of Kansas, Lawrence, KS, USA
[3]Marine Science Program, University of South Carolina, Columbia, SC, USA
[4]Department of Biological Sciences, University of South Carolina, Columbia, SC, USA
[5]Natural Resources Canada, Canadian Forest Service, Atlantic Forestry Centre, NL, Canada
[6]Present address: Environmental Science Division, Oak Ridge National Laboratory, Oak Ridge, TN, USA
[7]Present address: Great Lakes Institute of Environmental Research, University of Windsor, Windsor, ON, Canada

*Correspondence to*: Michael Philben (philbenmj@ornl.gov)

**Abstract.** Mosses contribute an average of 20% of upland boreal forest net primary productivity and are frequently observed to degrade slowly compared to vascular plants. If this is caused primarily by the chemically complexity of their tissues, moss decomposition could exhibit high temperature sensitivity (measured as $Q_{10}$) due to high activation energy, which would imply soil organic carbon (SOC) stocks derived from moss remains are especially vulnerable to decomposition with warming. Alternatively, the physical structure of the moss cell wall biochemical matrix could inhibit decomposition, resulting in low decay rates and low temperature sensitivity. We tested these hypotheses by incubating mosses collected from two boreal forests in Newfoundland, Canada, for 959 days at 5 and 18°C, while monitoring changes in the moss tissue composition using total hydrolysable amino acid (THAA) analysis and $^{13}$C NMR spectroscopy. Less than 40% of C was respired in all incubations, revealing a large pool of apparently recalcitrant C. The decay rate of the labile fraction increased in the warmer treatment, but the total amount of C loss increased only slightly, resulting in low $Q_{10}$ values (1.23-1.33) compared to L horizon soils collected from the same forests. NMR spectra were dominated by O-alkyl C throughout the experiment, indicating the persistence of potentially labile C. Accumulation of hydroxyproline (derived primarily from plant cell wall proteins) and aromatic C indicates selective preservation of biochemicals associated with the moss cell wall. This was supported by scanning electron microscope (SEM) images of the moss tissues, which revealed few changes in the physical structure of the cell wall after incubation. This suggests the moss cell wall matrix protected labile C from microbial decomposition, accounting for the low temperature sensitivity of moss decomposition despite low decay rates. Climate drivers of moss biomass and productivity, therefore, represent a potentially important regulator of boreal forest SOC responses to climate change that needs to be assessed to improve our understanding of carbon-climate feedbacks.

## 1 Introduction

Boreal forests account for over half of global forest soil carbon (C) stocks, with areal soil C densities 2-3 times higher than temperate or tropical forests (Malhi et al., 1999). The low temperatures and, in many regions, high





soil moisture likely cause slow rates of decomposition, contributing to the retention of these substantial stores of soil C (Coûteaux et al., 2002; Hobbie, 2013; Hobbie et al., 2000; Wetterstedt et al., 2010). However, these northern areas are expected to be vulnerable to climate change (IPCC, 2014). Warming impacts on carbon cycling and storage within boreal forests are likely due to changes in both C emissions from soils due to decomposition (Kane et al. 2005; Norris

et al. 2010) and changes in soil C sources even prior to major vegetation shifts associated with long-term climate change (Kohl et al. 2017; Ziegler et al. 2017).

       Mosses contribute an average of about 20% of the total NPP in upland boreal forests (Turetsky et al., 2010) and can locally exceed vascular plant NPP (Frolking et al. 1996; Gower et al. 2001). Despite this, the unique dynamics of moss biogeochemistry are not typically included in models of boreal forest C cycling, which could result in

considerable biases. For example, Bona et al. (2013) used a literature review of the rates of primary production and decomposition of upland mosses to estimate the range of moss C that could be stored in the soils of black spruce-dominated boreal forests, and the result (31-49% of total SOC) was comparable to the difference between modeled and observed C stocks for those forests. This demonstrates that accurately representing moss production and decomposition is essential for modeling the C cycle in moss-rich boreal forests. While many studies estimate moss

primary production, only three estimates of degradation rates were available for this meta-analysis, and none estimated the temperature sensitivity of moss decomposition (Bona et al. 2013). This is significant because the limited numbers of studies of upland mosses show slower decomposition than vascular plant litter incubated under similar conditions (Harden et al., 1997; Moore and Basiliko, 2006). Omitting moss-specific dynamics could therefore overestimate decomposition and underestimate C storage in moss-rich soils.

Predicting the temperature sensitivity of moss decomposition is also difficult because it is not clear if the apparent recalcitrance of moss tissues is due to chemical complexity, physical and structural characteristics impeding microbial decomposition, or a combination of the two. Most studies of moss biochemistry have focused on the peat-forming *Sphagnum* genus, and it is not clear to what extent these insights apply to non-*Sphagnum* upland species. While mosses do not contain lignin (Maksimova et al., 2013), they do produce a variety of lignin-like phenols, which

have been hypothesized to inhibit decomposition (Tsuneda et al., 2001). In addition, *Sphagnum* mosses produce structural carbohydrates that also appear to impede decomposition (Hájek et al., 2011; Turetsky et al., 2008). If the decomposition of upland mosses is limited by chemical properties, we might expect higher temperature sensitivity of moss decomposition compared to vascular plant litter due to higher activation energy of more chemically complex compounds (Bosatta and Ågren, 1999; Davidson et al., 2006). This result would suggest the moss C pool is not only

under-represented in size due to slow decomposition, but is also particularly vulnerable to decomposition with warming.

       Distinct from moss chemical characteristics, the physiochemical matrix of moss cell walls also could play a role in limiting microbial access to moss tissues otherwise useful as microbial resources. Indeed, scanning electron micrographs of a slowly decomposing *Sphagnum* species (*S. fuscum*) revealed little change in the structure of the cell

wall after three years of decomposition, suggesting that something inherent about moss cell wall structure presented a barrier to microbial access, even long after cell death (Turetsky et al. 2008). If so, moss tissue chemistry *per se* may not require a higher activation energy than vascular plant tissues for decay to proceed, but decay instead would be



limited by microbial physical accessibility to usable resources. However, it is not clear if the observed decay resistance is widespread, as the cell wall structure of another *Sphagnum* species (*S. riparium*) collapsed after one year of decomposition (Turetsky et al. 2008). The degree to which upland moss cell walls retain their physical structure and impede microbial decomposition remains unknown.

We observed the decay of upland boreal forest mosses collected from the Newfoundland and Labrador Boreal Ecosystem Latitudinal Transect (NL-BELT) for more than 2.5 y to investigate (1) the temperature sensitivity of moss tissue decomposition; and (2) the relationship between moss chemical composition, cell wall structure, and its decomposition. We combined chemical characterization with scanning electron microscopy (SEM) to determine both chemical and physical changes in the moss tissues during decomposition. In doing so we investigate the relative
importance of these factors contributing to the slow turnover of moss tissues in these forests.

## 2 Methods

### 2.1 Sample sites

Moss samples were collected in July 2011 from two balsam fir dominated forest sites from within the Newfoundland and Labrador Boreal Ecosystem Latitudinal Transect (NL-BELT). One site was located in the Salmon
River watershed near Main Brook on Newfoundland's northern peninsula (hereafter "SR"), and one in the Grand Codroy watershed in southwestern Newfoundland ("GC"). The mean annual temperature (MAT) at SR is 3.2°C lower than GC (2.0 vs. 5.2°C; Table 1), while mean annual precipitation is higher at GC (1224 vs. 1505 mm; Environment Canada Normals 1981-2010). The dominant moss species were *Dicranum spp.* and *Rhytidiadelfus spp.* in SR, and *Pluerozium spp., Hylocomium splendens*, and *Ptillium crista-cristensis* in GC, which had lower overall moss coverage
based on examination of 15 1 m$^2$ plots at each site (Table 1). Representative samples of the dominant moss species were collected from both sites, rinsed with DI water, and homogenized by site prior to incubation. We sampled mosses from multiple patches in both forests. The collection was focused on accurately reflecting the moss cover at each site and no effort was made to ensure all species were represented.

### 2.2 Incubation experiment

Incubations were designed to include four destructive-sampling time points which occurred at 69, 283, 648, and 959 days from the beginning of the experiment, starting in October 2011. Mosses from both regions were incubated in the dark within sealed glass jars at 5°C and 18°C. For each site, 24 replicate jars were established that contained 1, 1.5, 2.5, or 2.75 g dry weight of moss tissue; jars containing greater weights were sampled at later time points. A microbial inoculum derived from each site's organic horizon was added to their respective jars at the start
of the incubation experiment. A soil slurry was created by saturating the soil sample from each site with nanoUV water (~120 gdw soil L$^{-1}$). The slurry was then filtered (GF/C; 1.2 μm nominal pore size) to exclude most soil organic matter while retaining many soil microorganisms and pipetted onto the moss sample in each jar at a volume required to achieve 60% water holding capacity (6-21 mL for the 1-2.75 g dry weight moss samples). Following inoculation, jars were sealed and incubated, with half the jars at 5 and half at 18°C (12 jars for each site and temperature





combination). We opened all jars biweekly to allow for gas exchange and to add sufficient water to ensure moss was kept at approximately 60% water holding capacity. At each sampling time point all moss was removed from the jars and dried at 40°C to constant mass and weighed. Once weighed, moss was ground using a Wiley mill (Thomas Scientific, Swedesboro) with a 60 mesh (0.25 mm) screen and stored in glass vials in the dark for subsequent analyses.

**2.3 Chemical analyses**

To determine how decomposition affected organic matter composition, the elemental and stable carbon and nitrogen isotopic composition of the moss samples were analyzed on a Carlo Erba NA 1500 Series elemental analyzer interfaced to a Delta V Plus isotope ratio mass spectrometer via a ConfloIII interface (ThermoFisher Scientific). In total 48 samples were analyzed, plus 6 initial samples taken as random triplicates from the homogenized prepared
moss tissue from each site.

Sub-samples of the initial and final (959 days) moss samples were analyzed using solid state CPMAS $^{13}$C-NMR to determine the proportions of carbon functional groups and how they changed with decomposition. Samples were analyzed using a Bruker AVANCE II 600 MHz using a MASHCCND probe. All samples were run at 150.96 MHz for $^{13}$C and spun at 20 kHz at a constant temperature of 298 K. Carbonyl and amide (190-165 ppm) were
separated from each other by subtracting N:C (multiplied by 100%) from the total peak area ratio to determine the maximum amount carbonyl. The remaining provides an approximation of the maximum amide C proportion.

Water soluble inorganic nitrogen, measured as nitrate plus nitrite and ammonium, was determined using a Lachat 8500 flow injection analyzer. All samples were first extracted using NanoUV water. Briefly, 300 mg of the ground moss material were shaken with 10 ml of water for 2 minutes at room temperature then centrifuged and filtered
using a glass fiber filter (GF/F; nominal pore size of 0.45 μm) to remove particulate matter. The filtrate was then analyzed for $NO_3^-$ and $NH_4^+$ concentration to determine the total water soluble inorganic nitrogen content of each moss tissue sample.

Total hydrolysable amino acids (THAA) were analyzed following the method outlined in Philben et al. (2016) using the EZ:Faast kit for amino acid analysis (Phenomenex, USA). Briefly, 20mg subsample of each moss sample
was mixed with 1 ml of 6M HCl acid in a 1 ml ampule, which was sealed, shaken, and heated at 110°C for 20 hours. Samples were then transferred to 2 ml vials, and centrifuged. An aliquot of the resulting hydrosylate was transferred into a new vial and evaporated using $N_2$ gas. 200 μl of 0.01 M HCl was added to each hydrosylate along with norvaline which was added as an internal standard. Amino acids were derivatized with propyl chloroformate using the EZ:Faast kit. Samples were analyzed on an Agilent 6890 gas chromatograph with a ZB-AAA column using a single step oven
program of 110-320°C at 30° min$^{-1}$ and quantified using a flame ionization detector. Fifteen amino acids were determined; alanine, glycine, valine, leucine, isoleucine, threonine, serine, proline, aspartic acid, hydroxyproline, glutamic acid, phenylalanine, lysine, histidine and tyrosine.

**2.4 Scanning electron microscopy**

We performed scanning electron microscopy (SEM) on prepared fragments of the initial and final moss
tissues from this experiment using a JEOL JSM 7100F Field Emission SEM equipment with a Thermo EDS. Three





random subsamples where taken from the homogenized whole dried (40˚C) initial moss samples used to establish the incubation mesocosms. These were pooled to provide representative samples of the initial moss tissues. Subsamples of each of the final 18˚C incubation samples were used to provide the most degraded samples to compare with the initial sample images. All samples where dried at 50˚C, mounted on aluminum stubs, and coated with 300 angstroms

of gold using a SPI-Module Sputter Coater (Structure Probe, Inc.; West Chester, PA, USA).

## 2.5 Data analysis

Percent mass remaining was calculated using the dry weights of the initial and final mass at the four time points of the incubation. Mass remaining was then fit to the exponential decay equation to calculate the rate of decay:

$$y = L\left(e^{-kt}\right) + R. \tag{1}$$

$L$ represents the labile fraction of mass that was decomposed by the end of the experiment, $R$ was the residual fraction was left undecomposed, $t$ represents time in days and $k$ is the exponential decay constant (day$^{-1}$). Carbon and nitrogen remaining was also fit to equation (1) where possible.

Q$_{10}$ values were calculated to measure the temperature sensitivity of moss decomposition. This was performed by comparing C loss over the full experiment (equation 2), or using the k-values calculated in equation (1)

to estimate the temperature sensitivity of decomposition of the labile fraction (equation 3).

$$Q_{10,Total} = \left(\frac{CL_{T2}}{CL_{T1}}\right)^{\frac{10}{T2-T1}} \tag{2}$$

$$Q_{10,Labile} = \left(\frac{k_{T2}}{k_{T1}}\right)^{\frac{10}{T2-T1}} \tag{3}$$

CL indicates the percentage C loss, T2 indicates the warmer temperature (18°C), and T1 indicates the cooler temperature (5°C).

THAA data were analyzed to determine the percentages of total C or N as amino acids, and their change over incubation time, using the equation from Philben et al. (2016):

$$\text{THAA (\%C or N)} = \Sigma[\text{Yield}_{AA}/(\text{C or N})] \times [\text{Wt \% (C or N)}]_{AA} \tag{4}$$

where Yield$_{AA}$/(C or N) is the normalized yield of each amino acid (15 in total) in mg amino acid per 100mg C or N and [Wt % (C or N)]$_{AA}$ is the weight % of C or N in the amino acid. This equation was used for each individual amino

acid, which was then summed for each sample. The THAA results were also used to determine the degradation index, commonly used for interpreting the extent organic matter diagenesis in aquatic environments (Dauwe et al., 1999; Dauwe and Middelburg, 1998; Menzel et al., 2015). The index was modified for use at the sites used in the current study to permit assessment of the degradation state of the sites' soil organic matter pools (Philben et al. 2016). In that work, a principle component analysis (PCA) of a data set including green mosses, pooled needle litterfall, and L, F,

H and B soil horizons from three regions within the NL-BELT was conducted to identify the most significant differences in the overall THAA composition. Scores of the first principle declined with increasing decomposition. This approach allows for the comparison of the moss tissues in the current incubation study and the soil profiles at each site. The equation taken from Philben et al. (2016) was:

$$\text{Degradation Index} = \left[\Sigma_i\left(\frac{Mol_i - Avg_i}{SD_i}\right) \times PCI_i\right] \tag{5}$$





where $Avg_i$ and $SD_i$ are the average and standard deviation of the individual amino acid (mol%) determined for the data set described above, $Mol_i$ is the mol% of each amino acid analyzed from the mosses, and $PCl_i$ is the loading of the amino acids on the first principle component of the PCA performed on the dataset of all litter, soil horizons, and moss from all regions across the NL-BELT transect. See Philben et al. (2016) for more detail.

To test for effects of incubation temperature, site, and their interaction on all quantified variables, we applied two-way ANOVA using a mixed effects model to account for the pseudoreplication of repeated measures. Additional 2-way ANOVA tests were conducted within each time point to determine if the treatment effects changes over the course of the incubation. A two-way ANOVA was conducted to test the effects of site and temperature on Alkyl: O-Alkyl, using the initial moss samples and final (day 959) moss samples at 5 and 18°C, and the effects of site and time

(before or after incubation) on mol% hydroxyproline and the degradation index All other analyses were conducted in R 3.3.2 (R core team, 2016).

## 3   Results

### 3.1 Moss tissue decay rates and mass and elemental loss.

Both C and mass loss measured in this experiment were fit to an exponential equation as described in the
methods (Fig 1). When incubated at 18°C, between 34 and 38% of the initial moss was lost, based on both the mass and C loss results. The rate of loss in the 18˚C incubation declined over time, with rapid loss from 0-283 days, less change from 283-648 days, and no change from 648 to 959 days. In the 5°C incubations, between 28 and 30% of mass or C was lost, with mass and C losses continuing throughout the incubation. Regardless of incubation temperature, the percentage of mass or C remaining at the end of the 959 days was not different between sites (Table 2). Decay
constants (k) associated with moss tissue mass or C remaining did not differ between sites and were greater in the 18°C incubation than at 5 °C (-0.0130±0.0015 vs. -0.00380±0.0007 respectively; average ± standard deviation). The $Q_{10}$ values of mass loss during the full incubation ($Q_{10, total}$) averaged 1.24 and did not vary by site. The $Q_{10}$ of the labile pool calculated from the fitted k-values were 3.56 and 2.08 for SR and GC, respectively.

The N remaining did not follow the same trend as mass and C, and could not be fitted to an exponential curve
but rather exhibited both increases and decreases over the course of the incubation (Fig. 2). One treatment (GC at 18°C) experienced a net gain in total N during the incubation, suggesting N fixation occurred. Temperature did not have an effect on N remaining at most time points (Table 3). However, collection site had an effect on N remaining during decomposition at all time points except on day 69 due to greater N loss during the 18°C incubation of GC mosses (up to 46%; Table 2).

### 3.2 Elemental and stable isotope composition of decaying moss tissues.

Mosses collected from the two sites differed in initial N concentration (0.92±0.05% and 0.59±0.03% in the warmer (GC) and cooler (SR) forest site, respectively; t-test, p=0.003) but were not different in C concentration (43.9±0.1% and 43.6±0.5, respectively, p=0.373; Table 2, Table 3). This variation in N was responsible for the lower C:N ratio of 55.6±3 at GC relative to the higher value of 86.3±3 at SR (p<0.001). However, C:N declined with



decomposition to a greater extent in SR than GC, and the site difference in C:N decreased with incubation time (Fig. 3). Though the effect of site on C:N of the moss tissue was significant throughout the experiment, the differences between sites became less pronounced with time, especially in the 18°C incubations, and the p-value increased over the course of the experiment (Table 3).

The initial $\delta^{13}C$ and $\delta^{15}N$ values of moss did not differ between sites. $\delta^{13}C$ was -31.8±0.5‰ and -32.0±0.2‰ for SR and GC (p=0.3179), while $\delta^{15}N$ was -3.19±0.2‰ and -3.72±0.4‰, respectively (p=0.1679). Moss $\delta^{13}C$ increased over the course of the 18°C incubation for both sites (p=0.010) and increased during the 5°C only for GC. The increase in $\delta^{13}C$ was greatest in the 18°C incubation of GC mosses (1.5‰; Fig. 1). In all other incubations $\delta^{13}C$ increased by less than 1‰, with most of that change occurring within the first 400 days when the rate of C and mass

loss was highest. Therefore, although $\delta^{13}C$ of moss tissues was not different for the initial values collected, the $\delta^{13}C$ values differed by site throughout the decomposition experiment with the significance of that difference increasing with decomposition (Table 2). There was no effect of temperature on the $\delta^{13}C$, until the final time point when site, temperature, and their interaction had a significant effect due to an increase of $\delta^{13}C$ values in the GC 18°C incubation to -30.6±0.3‰. Temperature, but not site, had an effect on $\delta^{15}N$ (p=0.035; Table 2). $\delta^{15}N$ increased during incubation

in all treatments except for SR at 18°C, by an average of 2.3±1.5‰ (Figure 2).

### 3.3 Molecular composition of C and N and SEM images of decaying moss tissues.

Nitrogen was further characterized into four compound classes: nitrate+nitrite, ammonium, total hydrolysable amino acids, and molecularly unidentified N (MUN) and expressed as a fraction of total N content (Fig. 4). Water-extractable ammonium increased slightly with decomposition from initial values of 2-2.5% to 4-7% in the

18°C treatment and 5-11% in the 5°C treatment. Extractable nitrate+nitrite was less than 1% of total N in most samples, with the exception of the final time point when some nitrate values reached up to 20% of total N. Relatively high nitrate concentrations at the final time point were associated with elevated N remaining values (> 100%) and more negative $\delta^{15}N$ values (Fig. 2 and 4). THAA declined from 51.2±3 and 50.9±5% to 26.4±7 and 21.2±3% of the total N in SR and GC, respectively (p>0.001), and the effect of site was not significant on the decline (p=0.238). The

decline in the %N as THAA over the incubation coincided with changing amino acid composition, as the degradation index also declined as expected with decomposition (p>0.001). The change was larger in SR, declining from 1.4±0.7 and 2.6±0.5 to -1.6±0.5 and -1.3±0.1 in GC and SR, respectively (p=0.0388). Mole % hydroxyproline increased over the first 69 days in both sites (from 0.9 to 1.5% in GC and 1.1 to 1.3% in SR, Fig. 5). It remained elevated for the remainder of the incubation in GC and was significantly higher than the initial value (p=0.0496), but declined back to

1.1% in SR after 959 days. Because %N as THAA declined with little change in the inorganic N pools, MUN increased in relative abundance with decomposition in both regions.

The CPMAS $^{13}C$-NMR results indicate that the moss tissues from both forest sites were similar and dominated by O-alkyl and di-O-alkyl C (70%) with relatively little alkyl, carbonyl or aromatic C (Fig. 6; Table 3). The NMR spectra of the moss tissues before and after incubation were broadly similar and exhibited little change in

the relative proportions of the major C groups. As a result, the alkyl:O-alkyl ratio also exhibited no change with decomposition regardless of collection site and despite up to 50% mass loss. The largest change observed was in the





calculated maximum amide value from the carbonyl-C and amide-C resolved at chemical shift 190-165 ppm. Calculated maximum amide-C increased from $1.85\pm0.1\%$ to $2.30\pm0.04\%$ and $1.16\pm0.06\%$ to $2.18\pm0.1\%$ for mosses incubated at 18°C from SR and GC, respectively. Amino acids are likely a major source of amide C; however, the %C as THAA declined with decomposition, indicating the increasing relative abundance of amide C was due to MUN

compounds.

SEM imaging revealed few apparent differences between the physical structure of moss tissues before and after incubation (Fig. 7, S1, and S2). Intact moss cell walls were visible in both sets of images with few signs of structural change. There was no evidence of warping, gouging, or pitting from microbial degradation of the cell wall following the incubation of mosses from either region.

**4 Discussion**

**4.1 Decomposition of mosses is slower and less temperature sensitive than vascular plants**

The low decay rates observed are consistent with previous studies of moss decomposition (Fyles and McGill, 1986; Hobbie et al. 2000; Hogg, 1993; Hagemann and Moroni 2015). Decay rates of moss tissues are typically lower than rates for vascular plant decay under similar conditions (Fyles and McGill, 1987; Hagemann and Moroni, 2015;
Hobbie, 1996). Mass loss during one-year litter bag decomposition of balsam fir needles averaged 27% at SR and 35% at GC (K. Edwards, unpublished data). The needles therefore experienced similar mass loss in one year compared to the mosses after 959 days, indicating more rapid decomposition. The moss decay rates can also be compared to incubations of L horizon soils collected from these sites, which were comprised of ~66% partially decomposed balsam fir litter in ER and ~83% in GC according to visual inspection. Despite their more advanced state of degradation
compared to the fresh moss litter, the L horizon soils also experienced C losses similar to those experienced by the moss samples (30-45%) in a shorter time period (68 weeks). The low decay rates observed are unlikely to be an artefact of the laboratory approach, as Bengstton et al. (2016) compared field-based litter bag and laboratory incubation approaches using a common set of *Sphagnum* mosses and found the laboratory approach generally resulted in greater mass loss.

The SR mosses, but not the GC mosses, exhibited higher $Q_{10}$ than the bulk L horizon soil (Laganière et al., 2015; Podrebarac et al., 2016) and previous findings for vascular plant tissue decomposition (e.g. Fierer et al. 2005) based on the decay rate of the labile C fraction. However, the higher temperature only slightly increased the total C degraded after 959 days. The $Q_{10}$ value based on total mass loss was therefore lower than the L horizon soils. This indicates that the additional energy in the warmer treatment was not sufficient to induce additional decomposition,
suggesting that decomposition was not limited by activation energy, contrary to the C-quality temperature hypothesis (Bosatta and Ågren, 1999; Davidson et al., 2006). Warming and drying trends in the boreal regions that inhibit moss growth (Gower et al., 2001; Turetsky, 2003) could result in the formation of SOM comprised of a greater relative abundance of vascular plant tissues, and thus of SOM that is both more decomposable and more temperature sensitive. This is consistent with our observations of increasing temperature sensitivity of soil respiration at lower latitudes along
this boreal forest transect where moss inputs are reduced (Podrebarac et al. 2016).



**4.2 The cell wall matrix governs low decomposition rates and temperature sensitivities of decay**

The low $Q_{10}$ values for total mass loss suggest the low decay rates are not caused by chemical complexity or recalcitrance of bulk moss tissues, properties associated with high activation energies and correspondingly high temperature sensitivities of decay. This idea is supported by the $^{13}$C NMR data, which indicate that the moss OM is

rich in carbohydrates with little aromatic or alkyl C. The proportions of O-alkyl C in mosses were higher than vascular plant litter collected from the same study sites (55.3% vs. 37.0%; Kohl et al. 2018). Further, the OM composition did not change significantly following incubation, unlike the decomposition of vascular plant tissues and SOM in which O-alkyl C is often preferentially degraded and alkyl C increases in relative abundance (Baldock et al., 1997; Kögel-Knabner, 1997; Preston et al., 2009). The lack of change in the moss alkyl:O-alkyl ratio during decomposition does

not appear to result from low mass loss in these incubations, given the increase in this ratio after a similar amount of mass loss (approximately 40%) in the foliage of each of 10 tree species in Canadian boreal forests (Preston et al., 2009). In conjunction with the relatively low temperature sensitivities of decay, the high proportion of O-alkyl C and lack of change in the alkyl:O-alkyl ratio following decomposition suggests that something other than bulk chemical composition governs the relatively low decomposition rates observed during the moss incubations.

There is also no evidence that decay-inducing microorganisms were limited by N availability. Indeed, varying concentrations of moss tissue N and C:N ratios across sites were not related to changing decay rates. Lower C:N ratios are typically correlated with faster decay in both mosses (Aerts et al., 2001; Bragazza et al., 2007; Limpens and Berendse, 2003) and vascular plant tissues (Nadelhoffer et al. 1992; Hobbie et al. 1996) in boreal soils, suggesting N limitation of decomposition. The N content was significantly lower and C:N higher in SR compared to GC (86.3±1.8

and 53.0±3.1, respectively), which would exacerbate N limitation in SR moss tissues. Limpens et al. (2003) demonstrated that *Sphagnum* decay resulted in net N mineralization below the threshold C:N of 67. Our data are consistent with these results, as net N loss was observed for the GC moss with C:N < 67 but not from the SR mosses with higher C:N. However, the lack of difference in decomposition despite greater N availability in GC suggests that N availability was not the limiting factor for moss decay rates.

The changing composition of N during the incubations suggests that rapid turnover of the N pool could have reduced microbial N limitation. The %N as THAA declined from ~50% to ~20% in the first 69 days, while the amino acid degradation index declined from ~2 to ~-1 over this period. Total N declined by <15% and increases in inorganic N were <5% over this period, indicating the degraded organic N was mostly transformed to MUN. Previous studies indicate the accumulation of amino sugars derived from microbial residues could contribute to the MUN pool

(Tremblay and Benner, 2006). The decline in %N as THAA and the degradation index are similar in magnitude to the difference in composition between the L and the H horizon in these soils (Philben et al. 2016). This indicates extensive degradation and turnover of the relatively small moss N pool, perhaps explaining the apparent lack of microbial N limitation despite high C:N ratios. The rapid turnover of the N pool indicates uncoupling of the N and C cycles in these incubations, and suggests the recalcitrant cell wall matrix does not protect most of the moss protein and peptide

N pool, which appear to remain labile.

Despite the rapid turnover of the bulk amino acid pool, changing AA composition during the incubation indicates the cell wall matrix protected some proteins as well. Hydroxyproline is found in glycoproteins in the plant



cell wall, but lacks major microbial sources (Philben et al. 2013). The increase in Hyp relative to other amino acids in the incubations of the GC mosses therefore indicates selective preservation of these cell wall proteins. The SR incubations exhibit an increase in mol% Hyp, followed by a decline to near the pre-incubation value. This regional difference is likely due to the difference in bulk N dynamics: the decline in C:N during the SR incubations indicates microbial N immobilization, which would produce amino acids but not Hyp and lower its mol%. Dilution of Hyp by microbial protein synthesis appears to have counteracted selective preservation of cell wall glycoproteins at SR. These dynamics further support the importance of the cell wall matrix in organic matter stabilization during the incubations.

The persistence of the physical structure of the cell wall likely explains the observed combination of slow decomposition and its low temperature sensitivity. The SEM images indicate microbial access to chemically labile C is inhibited by the biochemistry and physical matrix of the moss cell wall. The lack of microbial access to otherwise reactive microbial resources appears to be a bottleneck to decomposition that is not alleviated by increasing temperature, explaining our observation of low $Q_{10}$ and a large O-alkyl C pool despite apparently recalcitrant litter.

Although they share a backbone of cellulose microfibrils, there are important chemical and structural differences between moss and vascular plant cell walls which could contribute to the persistence of the moss cell wall during the incubation (Roberts et al. 2012). Uronic acids (glucuronic acid and galacturonic acid) are more abundant in mosses than vascular plants (Popper and Fry 2003). The NMR spectra indicated O- and di-O alkyl functional groups account for ~70% of the moss C, but molecular analysis of *Sphagnum* found about half of that total in five aldoses (glucose, galactose, mannose, rhamnose, and fucose) (Philben et al. 2014), leaving a large pool of uncharacterized moss carbohydrates. The low ratio of O-alkyl to di-O-alkyl C (3.7 vs. >4 in vascular plants) suggests that uronic acids contribute to this pool. *Sphagnum* also produces a group of pectin-like carbohydates (termed "sphagnan") composed primarily of rhamnose, mannose, and galacturonic acid (Ballance et al., 2007). This mixture appears to be resistant to decomposition and/or possess antimicrobial properties (Hájek et al., 2011; Stalheim et al., 2009). While this fraction has only been identified in *Sphagnum*, uronic acid-enriched carbohydrates in these upland mosses could play a similar role.

The NMR spectra also indicate a substantial contribution of phenolic and aromatic C (7% of the total C), despite the lack of lignin. The relative abundance of this fraction increased during decomposition, indicating selective preservation. Tsunida et al. (2001) identified an amorphous phenolic coating on the outside of the moss cell wall, and proposed that a specialized fungal consortium is required to break it down. This is analogous to the accelerated degradation of lignin in wood by white rot fungi (Rice et al. 2006), and is a plausible explanation of the lack of cell wall degradation observed in the SEM images as well as the accumulation of phenolic and aromatic C. If key fungal species were excluded from the soil slurry used as a microbial inoculum in the incubations, or if incubation conditions were not favorable for their growth, then their absence could lead to the persistence of the cell wall's structural integrity.

Overall, our data suggest that some combination of inherent molecular resistance to decomposition and the molecular architecture of the cell wall matrix make it difficult for exoenzymes to access and catabolize. While these analyses can identify biochemical differences between moss and vascular plant cell walls, we cannot identify which of these differences specifically contribute to their apparent recalcitrance. This presents an intriguing avenue for future





research, as the persistence of the physio-chemical integrity of the moss cell wall matrix is likely important for maintaining the globally significant pools of moss-derived C in both peatlands and forested uplands.

### 4.3 Implications for the bioavailability of boreal forest SOM

The contrasting responses of the bulk C composition and the THAA-based indices during moss decomposition could complicate interpretation of such data sets in moss-rich boreal forest soils. Decomposition of vascular plant tissues is typically characterized by selective loss of carbohydrates. Indices such as the carbohydrate yield and the O-alkyl to alkyl ratio of $^{13}$C NMR spectra have therefore become useful and widely used indicators of the degree of SOM degradation albeit with some caution (Baldock et al., 1997). However, our results indicate the O-alkyl C of moss tissues is not selectively degraded, and O-alkyl:alkyl C did not change following incubation (Fig. 8). The contribution of relatively recalcitrant moss-derived structural carbohydrates could cause the O-alkyl:alkyl C ratio to underestimate diagenesis in moss-rich boreal forest soils (Fig. 8d). In contrast, the change in the THAA-based indices (%N as THAA and the degradation index) following incubation was similar to the change with depth in the organic horizon of these soils (Fig. 8a,c) and the change during incubation of the L horizon (Philben et al. 2016). These results demonstrate variations in biochemical composition among plant types can confound conventional geochemical interpretation of diagenetic indices, and underscore the value of using multiple independent indices for a holistic understanding of the degradation of different SOM components (Baldock et al., 1997).

Our measurements of the decay rates and $Q_{10}$ of mosses contrast with latitudinal trends of bulk SOM bioavailability observed along the NL-BELT. Despite having the highest contribution of mosses to the organic horizon, the coolest region in the transect had the highest specific soil respiration rate ($R_{10}$) and similar or lower $Q_{10}$ compared to the other regions (Laganière et al., 2015; Podrebarac et al., 2016). The slow decomposition of moss tissues therefore does not appear to slow decomposition in moss-rich organic soils. However, the trend in organic matter composition is consistent with moss influence, as the coolest region is enriched in O-alkyl C and depleted in methoxy and aromatic C compared to the other regions (Kohl et al. 2017). The larger proportion of labile C compounds in the cooler region soils is correlated with the bioreactivity of the bulk soil (indicated by $R_{10}$; Kohl et al. 2017), suggesting moss contributions of labile C override the effects of slow decomposition of moss tissues themselves. This implies that the physical protection afforded by the cell wall declined in importance over time within the soil profile. It is possible that a fungal species or consortium absent from the soil inoculum can efficiently degrade the cell wall phenolics *in situ*, enabling the broader microbial community to utilize the labile moss C. Physical processes associated with the burial of moss litter and incorporation into SOM could also contribute to a loss of cell wall integrity over time. This study points to the need to better understand the physical and biochemical mechanisms controlling moss tissue decay and its variations among upland forest species and microhabitats, as these factors and their interaction impose important controls on boreal forest soil C stocks and their response to changing climate.

### Competing interests

The authors declare they have no conflict of interest.



**Data availability**

All data are included in the manuscript tables and the supplement file.

**Acknowledgements**

We thank Jamie Warren for overseeing the maintenance of the moss incubations and for assisting with sample
5    processing and analysis. Julia Ferguson assisted with moss sample preparation and mesocosm setup and monitoring.
Alex Morgan also assisted with incubation monitoring. We also thank Celine Schneider for conducting the $^{13}$C NMR
analyses, Alison Pye for conducting the stable isotope analyses, Wanda Aylward for conducting the SEM imaging,
and Geert Van Biesen for assisting with the hydrolysable amino acid analyses. This research was funded by the
National Science and Engineering Councils of Canada Discovery Grants program and Strategic Project Grants
10   program (STP-397494-10 and STPGP-479224), Centre for Forest Science and Innovation of the Newfoundland and
Labrador Agrifoods Agency, Canada Research Chairs Programme, and the Canadian Forest Service of Natural
Resources Canada.



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





**Table 1. Study site characteristics including mean annual temperature (MAT) and mean annual precipitation (MAP).**

|  | Salmon River | Grand Codroy |
| --- | --- | --- |
| Climate station | Main Brook | Doyles |
| MAT(°C)[a] | 2.0±2.4 | 5.2±2.0 |
| MAP(mm)[a] | 1223.9 | 1504.6 |
| rainfall (mm)[a] | 708.8 | 1110.4 |
| Snowfall (cm)[a] | 515.0 | 394.2 |
| GSST[b] | 14.00 | 16.0 |
| Latitude | 51°15'21.28"N | 47°53'36.34"N |
| Longitude | 56° 8'17.76"W | 59°10'28.31"W |
| Elevation (m) | 14 | 13.10 |
| Dominant moss Species[c] | *Pluerozium spp., Hylocomium splendens, Ptillium crista-cristensis* | *Dicranum spp., Rhytidiadelfus spp.* |

[a] Canadian climate normals, 1981–2010, http://www.climate.weatheroffice.gc.ca/climate_normals/index_e.html. MAT, mean annual temperature; MAP, mean annual precipitation.

[b] Growing season soil temperature (GSST) averaged over July and August in 2010 based on data from Canadian Forest Service, Atlantic Forestry Centre, in Corner Brook, Newfoundland, Canada.

[c] Identified by Kate Buckeridge, KJ Min, Kate Edwards, Andrea Skinner, Amanda Baker, Danny Pink



12  **Table 2. Results from 2-way ANOVA tests of the effects of collection site and incubation temperature on mass, C and N remaining, molar C to N ratio (C:N), and stable C**
13  **($\delta^{13}$C) and N ($\delta^{15}$N) isotope of the moss tissues. Results provided for both the full experiment (using a repeated measures ANOVA) and for each incubation time  (using 2-**
14  **way ANOVA). Bolded values are significant according ($\alpha$= 0.05).**

| | | | p-values | |
| --- | --- | --- | --- | --- |
| Parameter | Days | Temperature | Region | Interaction |
| %Mass Remaining | Full experiment | **<0.001** | 0.742 | 0.083 |
| (log transformed) | 0 | na | na | na |
| | 69 | **<0.001** | **0.028** | 0.463 |
| | 283 | **<0.001** | 0.283 | **0.009** |
| | 648 | **<0.001** | **0.011** | 0.088 |
| | 959 | **0.007** | 0.191 | 0.476 |
| %C Remaining | Full experiment | **<0.001** | 0.908 | 0.291 |
| (log transformed) | 0 | na | na | na |
| | 69 | **<0.001** | 0.375 | 0.562 |
| | 283 | **<0.001** | 0.879 | **0.014** |
| | 648 | **<0.001** | 0.429 | 0.310 |
| | 959 | **0.001** | 0.388 | 0.568 |
| %N Remaining | Full experiment | **0.007** | **0.009** | 0.090 |
| (log transformed) | 0 | na | na | na |
| | 69 | 0.053 | 0.238 | 0.506 |
| | 283 | 0.157 | **0.031** | 0.729 |
| | 648 | **0.005** | **<0.001** | **0.010** |
| | 959 | 0.716 | **0.012** | 0.207 |
| C:N | Full experiment | 0.869 | **<0.001** | **0.008** |
| (log transformed) | 0 | na | **<0.001** | na |
| | 69 | 0.383 | **<0.001** | 0.715 |
| | 283 | 0.051 | **<0.001** | **0.026** |
| | 648 | 0.232 | **0.008** | **0.015** |
| | 959 | **0.008** | **0.006** | 0.124 |



| | | | | |
|---|---|---|---|---|
| $\delta^{13}C$ | Full experiment | **0.010** | **<0.001** | 0.235 |
| | 0 | na | 0.113 | na |
| | 69 | 0.818 | **0.012** | 0.788 |
| | 283 | 0.103 | **0.002** | 0.240 |
| | 648 | 0.052 | **<0.001** | 0.423 |
| | 959 | **0.002** | **<0.001** | **0.025** |
| $\delta^{15}N$ | Full experiment | **0.035** | 0.417 | 0.160 |
| | 0 | na | **0.028** | na |
| | 69 | 0.762 | 0.225 | 0.797 |
| | 283 | **<0.001** | 0.164 | 0.309 |
| | 648 | 0.742 | **0.032** | 0.189 |
| | 959 | 0.841 | 0.485 | 0.286 |





17  Table 3. The CPMAS $^{13}$C-NMR results given as percentages of total C resolved for each chemical shift site provided, molar C to N ratio (C:N), and weight % C and N of
18  the initial and final moss tissues from both sites and incubated at 5°C and 18°C top.  All values are provided as the mean ± on standard deviation of three replicates.

| | Salmon River (cooler forest) | | | | | | Grand Codroy (warmer forest) | | | | | |
| | Initial | | Final (5°C) | | Final (18°C) | | Initial | | Final (5°C) | | Final (18°C) | |
|---|---|---|---|---|---|---|---|---|---|---|---|---|
| Alkyl (50-0) | 9.08 | ± 0.2 | 8.23 | ± 0.6 | 8.57 | ± 2 | 9.88 | ± 0.5 | 8.66 | ± 0.2 | 7.57 | ± 0.7 |
| O-alkyl (90-65) | 55.31 | ± 0.3 | 54.01 | ± 1.0 | 55.36 | ± 3 | 54.75 | ± 1.6 | 54.98 | ± 0.5 | 57.09 | ± 0.3 |
| Di-O-alkyl (110-90) | 15.57 | ± 0.1 | 15.32 | ± 0.1 | 15.19 | ± 1 | 15.11 | ± 0.1 | 15.06 | ± 0.2 | 15.49 | ± 0.1 |
| Aromatic (145-110) | 4.44 | ± 0.2 | 7.18 | ± 0.1 | 6.55 | ± 0.02 | 5.05 | ± 1.9 | 6.05 | ± 0.004 | 5.64 | ± 0.9 |
| Phenolic (165-145) | 2.35 | ± 0.2 | 2.07 | ± 0.1 | 1.74 | ± 0.2 | 1.68 | ± 0.4 | 1.57 | ± 0.1 | 1.24 | ± 0.01 |
| Carbonyl and amide (190-165) | 3.40 | ± 0.2 | 4.76 | ± 0.2 | 4.61 | ± 1 | 4.15 | ± 0.1 | 5.12 | ± 0.2 | 4.69 | ± 0.2 |
| Aromatic and Phenolic (145-110) | 6.79 | ± 0.4 | 9.25 | ± 0.1 | 8.29 | ± 0.2 | 6.74 | ± 2.3 | 7.62 | ± 0.1 | 6.87 | ± 0.9 |
| O-alkyl:Di-O-Alkyl | 3.55 | ± 0.03 | 3.53 | ± 0.04 | 3.64 | ± 0.1 | 3.62 | ± 0.1 | 3.65 | ± 0.01 | 3.69 | ± 0.002 |
| Alkyl:O-Alkyl | 0.16 | ± 0.003 | 0.15 | ± 0.01 | 0.16 | ± 0.04 | 0.18 | ± 0.004 | 0.16 | ± 0.002 | 0.13 | ± 0.01 |
| Carboxyl/ester * | 2.24 | ± 0.1 | 3.14 | ± 0.2 | 2.43 | ± 1 | 2.30 | ± 0.1 | 2.92 | ± 0.5 | 2.39 | ± 0.1 |
| Amide* | 1.16 | ± 0.06 | 1.62 | ± 0.4 | 2.18 | ± 0.1 | 1.85 | ± 0.1 | 2.20 | ± 0.2 | 2.30 | ± 0.04 |
| % Carbon | 43.6 | ± 0.5 | 44.4 | ± 0.4 | 43.3 | ± 0.3 | 43.9 | ± 0.1 | 44.1 | ± 0.2 | 42.3 | ± 0.7 |
| % Nitrogen | 0.590 | ± 0.03 | 0.823 | ± 0.1 | 1.08 | ± 0.07 | 0.920 | ± 0.05 | 1.12 | ± 0.09 | 1.19 | ± 0.08 |
| C:N | 86.3 | 3 | 64.0 | 9.7 | 46.9 | 2.7 | 55.6 | 3.2 | 46.1 | 3.7 | 41.6 | 3.4 |
| δ$^{15}$N | -3.19 | ± 0.2 | -1.25 | ± 0.8 | -2.42 | ± 0.4 | -3.72 | ± 0.4 | -1.61 | ± 2 | -0.8 | ± 2 |
| δ$^{13}$C | -31.8 | ± 0.5 | -32.1 | ± 0.2 | -31.8 | ± 0.2 | -32.0 | ± 0.2 | -31.5 | ± 0.03 | -30.6 | ± 0.3 |
| % N as Nitrate | 0.058 | ± 0.006 | 6.5 | ± 9.0 | 10.4 | ± 9 | 0.034 | ± 0.003 | 7.27 | ± 11 | 6.21 | ± 10.0 |
| % N as Ammonia | 2.41 | ± 0.1 | 8.41 | ± 2.0 | 5.24 | ± 2 | 1.59 | ± 0.06 | 8.14 | ± 3 | 6.99 | ± 1.0 |
| %C as amino acids | 2.54 | ± 0.03 | na | | 2.31 | ± 0.7 | 3.83 | ± 0.03 | na | | 2.12 | ± 0.7 |
| %N as amino acids | 52.1 | ± 1.1 | na | | 26.4 | 6.6 | 50.9 | ± 5 | na | | 21.2 | ± 3.0 |
| Total amino acids (nmol/mg) | 207.2 | ± 3 | na | | 193.0 | ± 60 | 313.0 | ± 30 | na | | 171.0 | ± 20.0 |

19  *maximum estimated value for amide-C and carboxyl/ester-C determined from the carbonyl-C and C:N ratio, see methods for details.

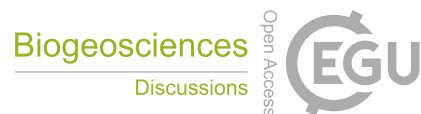

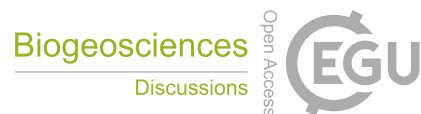

**Figure 1. Moss tissue C remaining normalized to the initial tissue C (black circles). The solid black line represents the exponential fit used to assess the decay constant (k). The δ¹³C values for the moss tissue at each time are indicated by squares and dashed lines, reported as the mean (n=3) with the standard error depicted by error bars. Panels a and c (blue lines and symbols) depict results of the 5°C incubation, and panels b and d (red lines and symbols) depict the 18°C incubation.**



**Figure 2.** **Percentage of initial nitrogen remaining (circles and solid lines) and δ¹⁵N (squares and dashed lines) at each time point. Points indicate the mean (n=3) with standard error depicted by error bars. Values in blue (panels a and c) are results from the 5˚C incubation and those in red (panels b and d) are from the 18˚C incubation.**



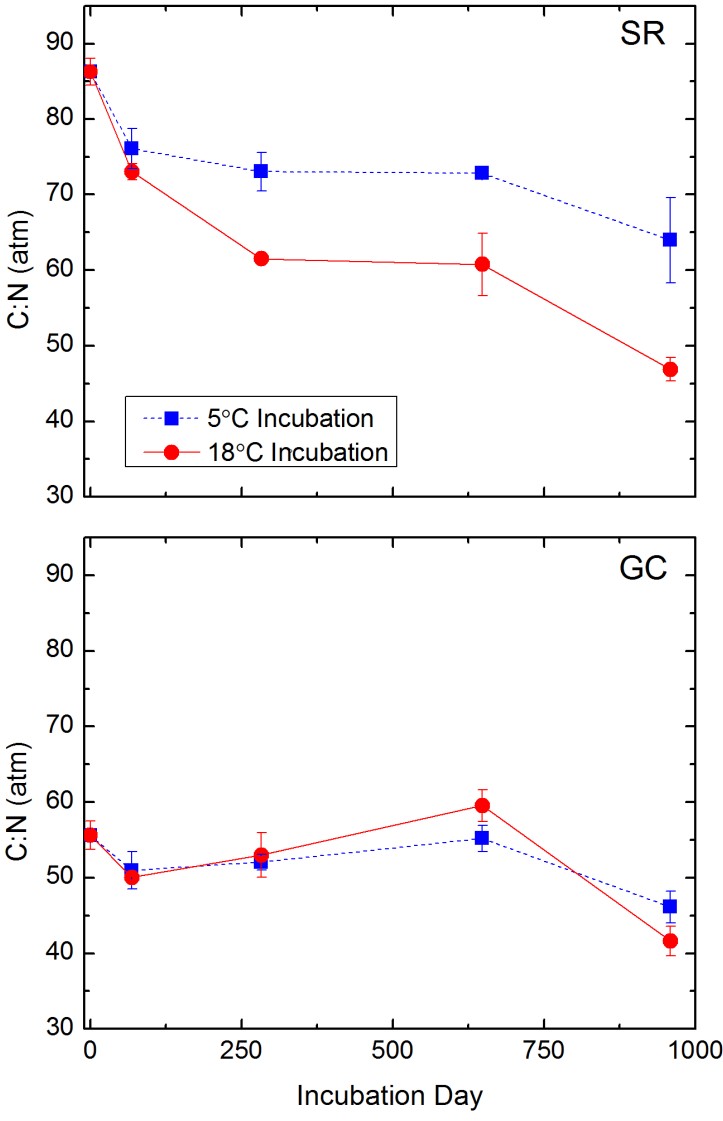

**Figure 3. Molar carbon to nitrogen ratio of moss tissues from the cooler (upper panel) and warmer (lower panel) forest sites plotted against incubation time. Values for the 5°C and 18°C incubations temperatures are given by the blue squares and red circles, respectively, and reported as the mean (n=3) with standard error depicted by error bars.**

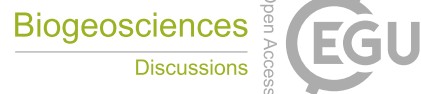

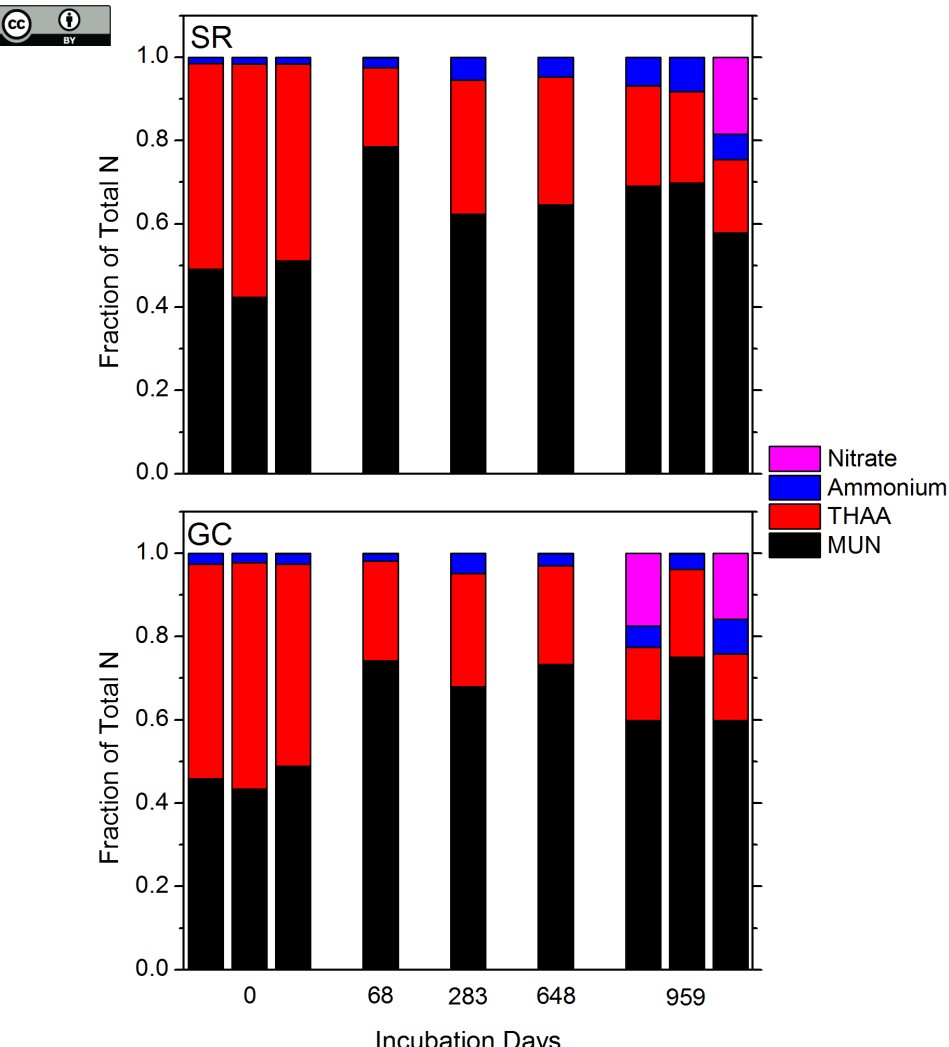

**Figure 4. Fraction of the total moss tissue N content as nitrate, ammonium, total hydrolysable amino acids (THAA), and molecularly unidentified N (MUN) in each sample. Total hydrolyzable amino acids were only reported for one triplicate of the 18°C incubations for the middle time points (69, 283, 648 days) though all samples were tested for the inorganic N species.**



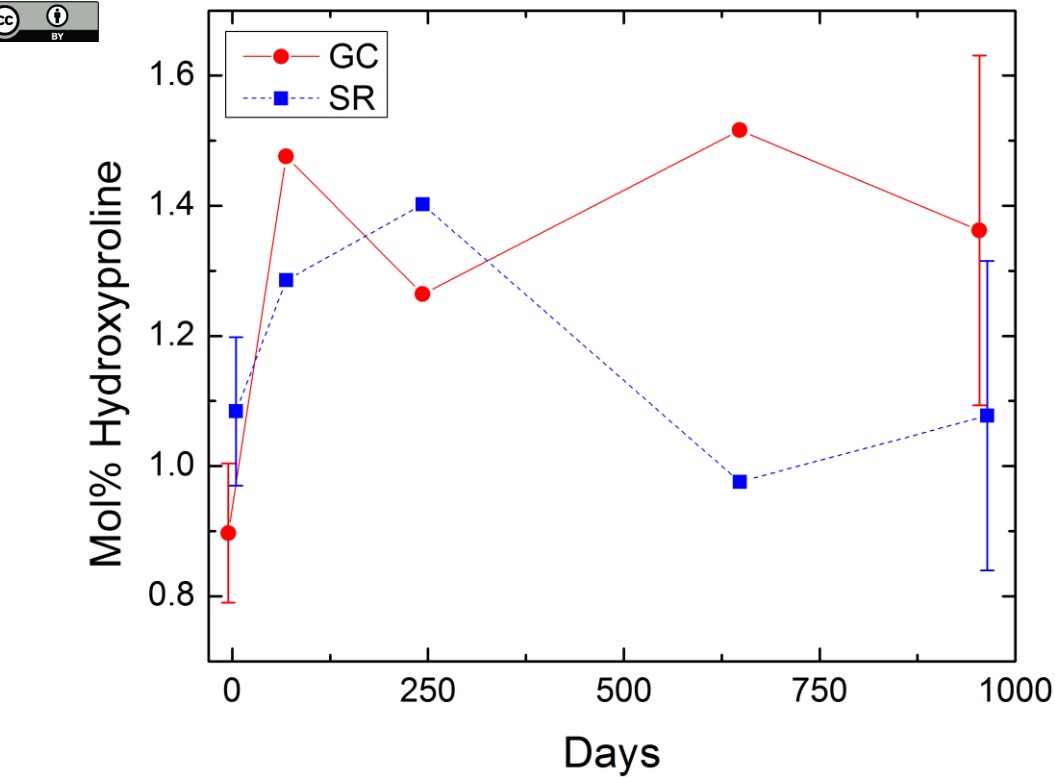

**Figure 5: Hydroxyproline yields in the moss tissues as a percentage of total hydrolysable amino acids. Error bars for the initial and final time points (0 and 959 days) indicate standard deviation (n=3). The initial and final time points are jittered (± 5 days) to display error bars without overplotting.**



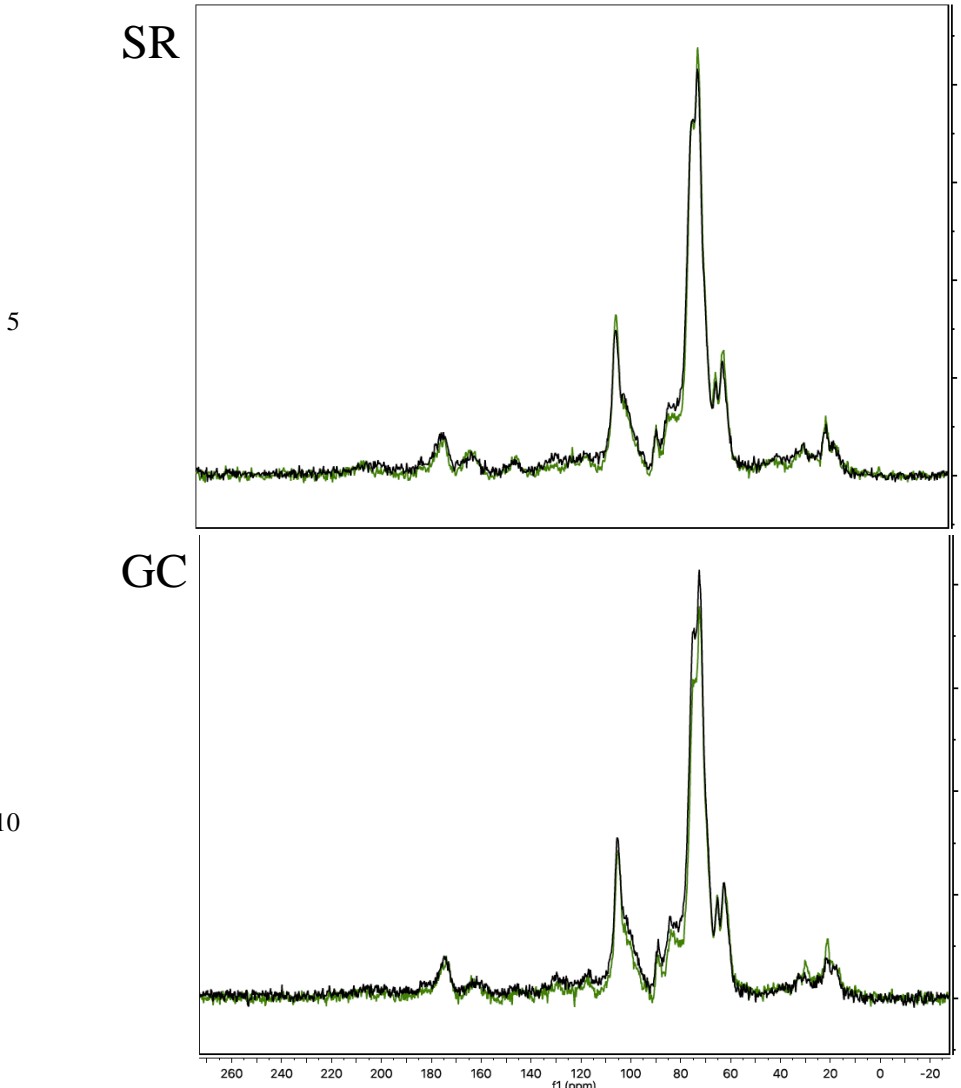

**Figure 6. Examples of the CPMAS $^{13}$C-NMR results for four out of twelve samples analyzed. The four chosen here are from each of the two forest sites with one undecomposed initial moss tissue replicate and the corresponding most decomposed replicate from the final time point of the 18°C incubation. Top panel provides examples from the cooler forest site (SR) and bottom panel the warmer forest site (GC). The initial samples are in green and a final sample from the 18°C incubation in black. The C types and their ppm range are; alkyl (50-0), O-alkyl (90-65), Di-O-alkyl (110-90), aromatic (165-110), carbonyl and amide (190-165).**



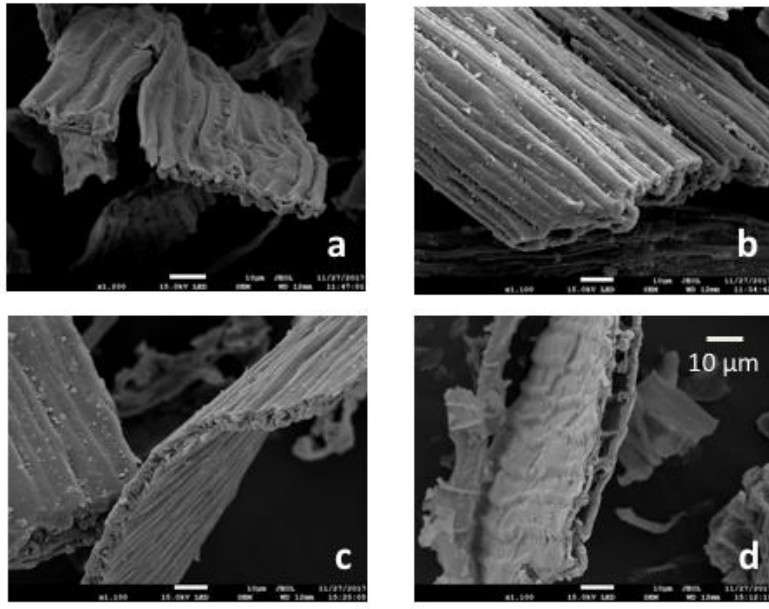

**Figure 7. Representative scanning electron micrographs of moss tissues before (a, c) and after (b, d) incubation. The top panels (a and b) depict mosses from the cooler forest, and the bottom panels (c and d) depict mosses from the warmer forest.**




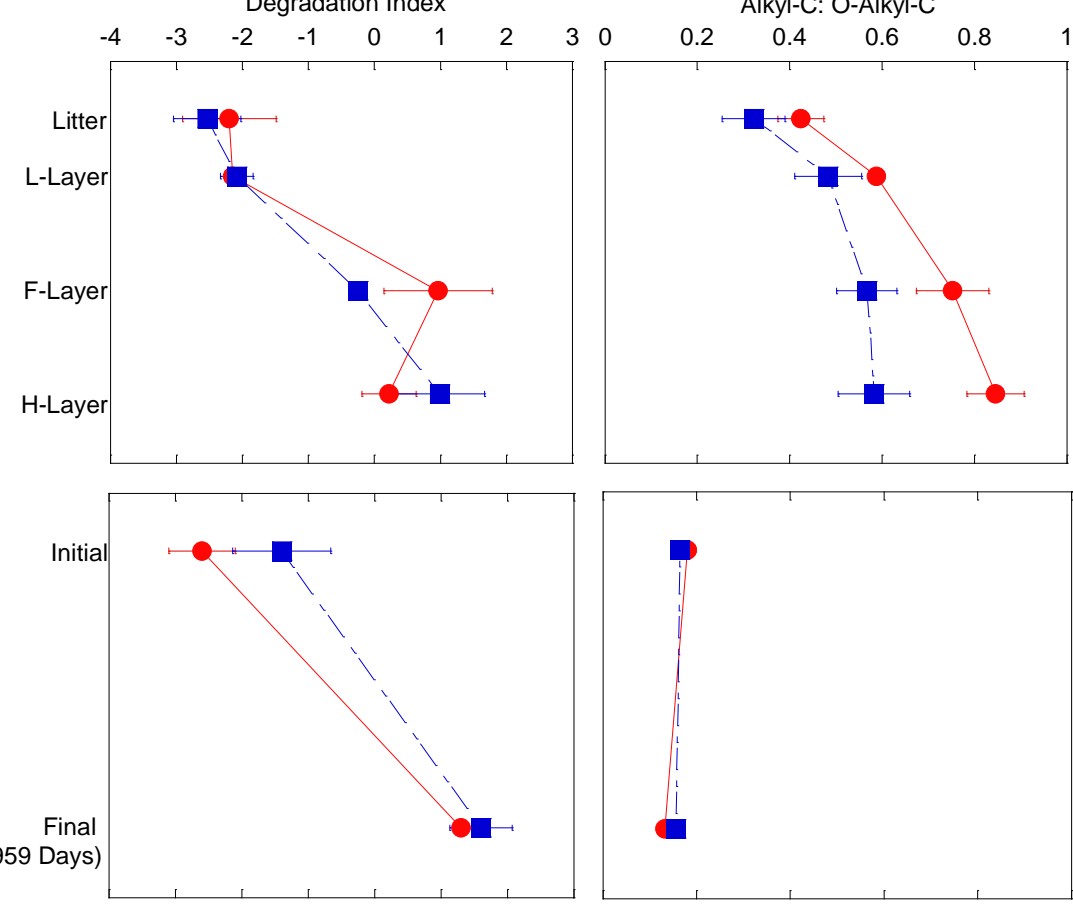

**Figure 8. Top two panels show the degradation index and Alkyl-C:OAlkyl-C ratio for the soil profile in the cooler region (blue squares) and warmer region ( red circles). Degradation index data from Philben et al (2016). Bottom two panels are calculated from this incubation data to contrast two methods for determining level of degradation for mosses. Data points are given as the mean with standard deviation displayed as error bars.**