# Peer review of "Biochemical and structural controls on the decomposition dynamics of boreal upland forest moss tissues"

_Biogeosciences, 2018_

## Referee Comment (RC1) · Anonymous Referee #1 · 3 Jul 2018

General comments The authors tested experimentally whether the slow degradability of boreal forest mosses is caused primarily by the chemically complexity of their tissues or the physical structure of the moss cell wall biochemical matrix inhibiting decomposition. The authors used various methods to study the decay rate of mosses, and changes in moss tissue C and N composition and physical structure during the 2.5-year laboratory incubation at two different temperatures. The results suggested 1) the moss cell wall matrix protected labile C from microbial decomposition and 2) the N and C cycles were uncoupled. I find the manuscript very interesting and topical in terms of assessing the role of boreal forest soils as sinks and sources of C. Below comments to the aspects listed by BG: 1. Does the paper address relevant scientific

questions within the scope of BG? YES. 2. Does the paper present novel concepts, ideas, tools, or data? YES. 3. Are substantial conclusions reached? YES. 4. Are the scientific methods and assumptions valid and clearly outlined? YES. 5. Are the results sufficient to support the interpretations and conclusions? YES. 6. Is the description of experiments and calculations sufficiently complete and precise to allow their reproduction by fellow scientists (traceability of results)? YES. 7. Do the authors give proper credit to related work and clearly indicate their own new/original contribution? YES. 8. Does the title clearly reflect the contents of the paper? YES. 9. Does the abstract provide a concise and complete summary? YES. 10. Is the overall presentation well-structured and clear? YES. 11. Is the language fluent and precise? YES. 12. Are mathematical formulae, symbols, abbreviations, and units correctly defined and used? YES. 13. Should any parts of the paper (text, formulae, figures, tables) be clarified, reduced, combined, or eliminated? YES, see specific comments. 14. Are the number and quality of references appropriate? I CANNOT ASSES THIS BECAUSE AT LEAST 14 REFERENCES GIVEN IN THE TEXT ARE MISSING FROM THE LIST OF REFERENCES. THE REFERENCES IN TEXT AND IN THE LIST SHOULD ALSO BE CROSS-CHECKED BECAUSE THERE ARE DIFFERENCES IN THE PUBLICATION YEAR OR NAME OF THE FIRST AUTHOR IN SOME CASES. 15. Is the amount and quality of supplementary material appropriate? YES.

Specific comments Page 3, line 13: Tell whether you only sampled green living (fresh?) parts of mosses or was the material a mixture of green and older brown parts.

Page 9, lines 33-35: Uncoupling of the N and C cycles has also been reported as a result of in situ incubations - see Manninen et al. 2016, Science of the Total Environment 571, 314-322. Add reference.

Technical corrections (typing errors, etc.) Page 3, lines 18-19 and Table 1: Correct the names of the moss species, i.e. should be Rhytidiadelphus spp., Pleurozium spp. and Ptilium crista-castrensis.

Page 6, line 27: I think the authors should refer to Table 2 (not Table 3).

Page 7, line 15: Replace 'Figure 2' with Fig. 2.

Page10, lines 25-33 (and page 11, lines 27-28): Discussion on fungi is very important, given that fungi are important decomposers in acid forest soils. If the authors have data on soil pH at the two sites, it should be added in Table 1.

Table 3: Replace '%Carbon' and '%Nitrogen' with %C and %N, respectively. Replace 'Nitrate' and 'Ammonia' with nitrate+nitrite and ammonium. Use nitrate+nitrite also on page 7, lines 21-22.

Fig. 8: Add a, b, c and d to indicate Figs. 8a-8d.

Cross-check the list of references with references in the text and revise when needed.

---

## Referee Comment (RC2) · Anonymous Referee #2 · 27 Jul 2018

This is an interesting and novel paper that tries to dig into the reasons behind the relatively well documented low decomposition rates of bryophytes that has a huge impact on biogeochemical cycles in the boreal, which as the authors point out, is frequently not taken into account. I think the question that this paper is addressing is important and novel.

I have a few concerns about the paper however that in my opinion would have to be addressed before it should be published. 1) The bryophyte species were not included as a variable in this test, but they did vary between regions. Bryophyte species, even beyond the true moss/sphagnum split, are far from being homogeneous. I suspect that

many of the differences found between the two regions has to do with the different species that were included in the mesocosmes. This point is not addressed anywhere in the text. There is considerable litterature showing that the nature of the decomposing matter is one if not the most important factor in determining decomposition rate (e.g. Lang et al. 2009 Journal of Ecology). Unfortunately the latin names of almost all the species are mispelled. I feel that including acknowldging this factor and including the associated litterature will strengthen this paper considerably. 2) The methods are not clearly enough described. In the annotated manuscript I have highlighted several places where more details are needed to clearly understand the methodology - mostly in the field aspects. Similarly, I am uncertain about the use of the Philben et al. 2006 approach as "green moss" from a stream is taken as equivalent as a variety of mosses from boreal forests. Can more justification be provided? 3) The results could be more clearly presented. I am uncomfortable with a table made up only of p values. It would be much better to have F values and N for the different tests. There also seems to be a contrediction between the table (effect of temperature on mass remaining), the figures (not really) and the texte (there was none). Also the figures were not always clear as information was lacking from the legends. I do wonder if all of the figures are required, perhaps Fig 6 could be an annexe?

In conclusion I think this is an interesting paper with a lot of potnetial. With a little refinement I think it could have a lot of impact.

Please also note the supplement to this comment:
https://www.biogeosciences-discuss.net/bg-2018-290/bg-2018-290-RC2-supplement.pdf

**Supplement:**

[revised manuscript text omitted]

---

## Author Comment (AC1) · 6 Sep 2018

**Response to interactive comments of Reviewer 1 (bg-2018-290)**

We thank reviewer 1 for helpful comments and corrections. Our responses to specific comments (reprinted in bold) are given below.

**The authors tested experimentally whether the slow degradability of boreal forest mosses is caused primarily by the chemically complexity of their tissues or the physical structure of the moss cell wall biochemical matrix inhibiting decomposition. The authors used various methods to study the decay rate of mosses, and changes in moss tissue C and N composition and physical structure during the 2.5-year laboratory incubation at two different temperatures. The results suggested 1) the moss cell wall matrix protected labile C from microbial decomposition and 2) the N and C cycles were uncoupled. I find the manuscript very interesting and topical in terms of assessing the role of boreal forest soils as sinks and sources of C. Below comments to the aspects listed by BG: 1. Does the paper address relevant scientific C1 BGD Interactive comment Printer-friendly version Discussion paper questions within the scope of BG? YES. 2. Does the paper present novel concepts, ideas, tools, or data? YES. 3. Are substantial conclusions reached? YES. 4. Are the scientific methods and assumptions valid and clearly outlined? YES. 5. Are the results sufficient to support the interpretations and conclusions? YES. 6. Is the description of experiments and calculations sufficiently complete and precise to allow their reproduction by fellow scientists (traceability of results)? YES. 7. Do the authors give proper credit to related work and clearly indicate their own new/original contribution? YES. 8. Does the title clearly reflect the contents of the paper? YES. 9. Does the abstract provide a concise and complete summary? YES. 10. Is the overall presentation well structured and clear? YES. 11. Is the language fluent and precise? YES. 12. Are mathematical formulae, symbols, abbreviations, and units correctly defined and used? YES. 13. Should any parts of the paper (text, formulae, figures, tables) be clarified, reduced, combined, or eliminated? YES, see specific comments.**

**14. Are the number and quality of references appropriate? I CANNOT ASSES THIS BECAUSE AT LEAST 14 REFERENCES GIVEN IN THE TEXT ARE MISSING FROM THE LIST OF REFERENCES. THE REFERENCES IN TEXT AND IN THE LIST SHOULD ALSO BE CROSS-CHECKED BECAUSE THERE ARE DIFFERENCES IN THE PUBLICATION YEAR OR NAME OF THE FIRST AUTHOR IN SOME CASES.**

Discrepancies between the reference list and the references cited in the text will be corrected in the revised manuscript.

**15. Is the amount and quality of supplementary material appropriate? YES.**

**Page 3, line 13: Tell whether you only sampled green living (fresh?) parts of mosses or was the material a mixture of green and older brown parts.**

The collected mosses were separated into green and brown fractions, and the green tissues were used in the incubations. This will be clarified in the revised manuscript.

**Page 9, lines 33-35: Uncoupling of the N and C cycles has also been reported as a result of in situ incubations - see Manninen et al. 2016, Science of the Total Environment 571, 314-322. Add reference.**

This reference will be added in the revised manuscript

**Page 3, lines 18-19 and Table 1: Correct the names of the moss species, i.e. should be Rhytidiadelphus spp., Pleurozium spp. and Ptilium crista-castrensis.**

Will be corrected in the revised manuscript

**Page 6, line 27: I think the authors should refer to Table 2 (not Table 3).**

Will be changed to Table 2 in the revised manuscript

**Page 7, line 15: Replace 'Figure 2' with Fig. 2.**

Will be corrected in the revised manuscript

**Page10, lines 25-33 (and page 11, lines 27-28): Discussion on fungi is very important, given that fungi are important decomposers in acid forest soils. If the authors have data on soil pH at the two sites, it should be added in Table 1.**

The soil pH of the two sites is quite low (<4.5 in all cases) and will be added to Table 1

**Table 3: Replace '%Carbon' and '%Nitrogen' with %C and %N, respectively. Replace 'Nitrate' and 'Ammonia' with nitrate+nitrite and ammonium. Use nitrate+nitrite also on page 7, lines 21-22.**

Will be corrected in the revised manuscript

**Fig. 8: Add a, b, c and d to indicate Figs. 8a-8d.**

Will be corrected in the revised manuscript

**Cross-check the list of references with references in the text and revise when needed.**

Will be corrected in the revised manuscript

---

## Author Response (AR1)

**Response to reviewers' comments (bg-2018-290)**

Response to interactive comments of Reviewer 1

5    We thank reviewer 1 for helpful comments and corrections. Our responses to specific comments (reprinted in bold) are given below.

**The authors tested experimentally whether the slow degradability of boreal forest mosses is caused primarily by the chemically complexity of their tissues or the physical structure of the moss cell wall biochemical matrix**
10   **inhibiting decomposition. The authors used various methods to study the decay rate of mosses, and changes in moss tissue C and N composition and physical structure during the 2.5-year laboratory incubation at two different temperatures. The results suggested 1) the moss cell wall matrix protected labile C from microbial decomposition and 2) the N and C cycles were uncoupled. I find the manuscript very interesting and topical in terms of assessing the role of boreal forest soils as sinks and sources of C. Below comments to the aspects**
15   **listed by BG: 1. Does the paper address relevant scientific C1 BGD Interactive comment Printer-friendly version Discussion paper questions within the scope of BG? YES. 2. Does the paper present novel concepts, ideas, tools, or data? YES. 3. Are substantial conclusions reached? YES. 4. Are the scientific methods and assumptions valid and clearly outlined? YES. 5. Are the results sufficient to support the interpretations and conclusions? YES. 6. Is the description of experiments and calculations sufficiently complete and precise to**
20   **allow their reproduction by fellow scientists (traceability of results)? YES. 7. Do the authors give proper credit to related work and clearly indicate their own new/original contribution? YES. 8. Does the title clearly reflect the contents of the paper? YES. 9. Does the abstract provide a concise and complete summary? YES. 10. Is the overall presentation well structured and clear? YES. 11. Is the language fluent and precise? YES. 12. Are mathematical formulae, symbols, abbreviations, and units correctly defined and used? YES. 13.**
25   **Should any parts of the paper (text, formulae, figures, tables) be clarified, reduced, combined, or eliminated? YES, see specific comments. 14. Are the number and quality of references appropriate? I CANNOT ASSES THIS BECAUSE AT LEAST 14 REFERENCES GIVEN IN THE TEXT ARE MISSING FROM THE LIST OF REFERENCES. THE REFERENCES IN TEXT AND IN THE LIST SHOULD ALSO BE CROSS-CHECKED BECAUSE THERE ARE DIFFERENCES IN THE PUBLICATION YEAR OR NAME OF THE**
30   **FIRST AUTHOR IN SOME CASES.**

Discrepancies between the reference list and the references cited in the text have been corrected in the revised manuscript.

35   **15. Is the amount and quality of supplementary material appropriate? YES.**

**Page 3, line 13: Tell whether you only sampled green living (fresh?) parts of mosses or was the material a mixture of green and older brown parts.**

The collected mosses were separated into green and brown fractions, and the green tissues were used in the incubations. This has been clarified in the revised manuscript. Please see pg. 3 lines 23-29 of the methods section.

**Page 9, lines 33-35: Uncoupling of the N and C cycles has also been reported as a result of in situ incubations - see Manninen et al. 2016, Science of the Total Environment 571, 314- 322. Add reference.**

This reference has been added in the revised manuscript (Page 10 line 9)

**Page 3, lines 18-19 and Table 1: Correct the names of the moss species, i.e. should be Rhytidiadelphus spp., Pleurozium spp. and Ptilium crista-castrensis.**

Has been corrected in the revised manuscript

**Page 6, line 27: I think the authors should refer to Table 2 (not Table 3).**

Changed to Table 2

**Page 7, line 15: Replace 'Figure 2' with Fig. 2.**

Corrected

**Page10, lines 25-33 (and page 11, lines 27-28): Discussion on fungi is very important, given that fungi are important decomposers in acid forest soils. If the authors have data on soil pH at the two sites, it should be added in Table 1.**

The soil pH of the two sites is quite low (<4.5 in all cases) and has been added to both the methods section (Page 3, lines 19-20) and Table 1

**Table 3: Replace '%Carbon' and '%Nitrogen' with %C and %N, respectively. Replace 'Nitrate' and 'Ammonia' with nitrate+nitrite and ammonium. Use nitrate+nitrite also on page 7, lines 21-22.**

Corrected

**Fig. 8: Add a, b, c and d to indicate Figs. 8a-8d.**

Will be added in the revised manuscript

**Cross-check the list of references with references in the text and revise when needed.**

Corrected

Response to interactive comments of Reviewer 2

We thank reviewer 2 for helpful comments. Our responses to specific comments (reprinted in bold) are given below.

15 **This is an interesting and novel paper that tries to dig into the reasons behind the relatively well documented low decomposition rates of bryophytes that has a huge impact on biogeochemical cycles in the boreal, which as the authors point out, is frequently not taken into account. I think the question that this paper is addressing is important and novel. I have a few concerns about the paper however that in my opinion would have to be addressed before it should be published.**

**1) The bryophyte species were not included as a variable in this test, but they did vary between regions. Bryophyte species, even beyond the true moss/sphagnum split, are far from being homogeneous. I suspect that many of the differences found between the two regions has to do with the different species that were included in the mesocosmes. This point is not addressed anywhere in the text. There is considerable**

25 **litterature showing that the nature of the decomposing matter is one if not the most important factor in determining decomposition rate (e.g. Lang et al. 2009 Journal of Ecology). Unfortunately the latin names of almost all the species are mispelled. I feel that including acknowldging this factor and including the associated litterature will strengthen this paper considerably.**

30 We acknowledge that the study prevents the separation of effects of different moss species vs. regional effects due to the differences in moss species between sites. However, the main conclusions of the paper (low decomposition and $Q10$, little change in chemical composition or physical structure) arise from similarities between the two sites. The observation of these similarities despite contrasting climate, moss species, and N availability strengthens these conclusions. This is clarified in the discussion of the revised manuscript.

The main difference we observe between the sites was in N dynamics, including changes in %N remaining, C:N, and amino acids. We maintain that these differences most likely arise from higher moss N concentrations at GC than

SR because the changes are consistent with differences in N availability. The differences in moss N concentrations are likely due to site differences in N availability rather than species-specific differences because N concentrations of balsam fir needles follow the same pattern as the moss tissues (Ziegler et al. 2017). We expanded the section on differences in N dynamics between the site to acknowledge the possibility of moss species effects, including additional citations. Please see pg. 9 lines 17-25 and pg. 10 lines 4-16.

The spelling of the Latin names is corrected in the revised manuscript

**2) The methods are not clearly enough described. In the annotated manuscript I have highlighted several places where more details are needed to clearly understand the methodology - mostly in the field aspects. The details highlighted in the annotated manuscript will be clarified in the revision. Similarly, I am uncertain about the use of the Philben et al. 2006 approach as "green moss" from a stream is taken as equivalent as a variety of mosses from boreal forests. Can more justification be provided?**

Clarification of methodological details, particularly the field collections, are now included as requested. Please see p. 3 lines 20-29 and p.4 lines 6-20.

"Green mosses" in Philben et al. 2016 refer to the green portion of upland boreal forest moss tissues, separated from the underlying brown portion which was reported separately. The mosses in Philben et al. 2016 were collected from the same two forest sites as the present study and the same set of dominant species are represented. This is clarified in the revised manuscript

**3) The results could be more clearly presented. I am uncomfortable with a table made up only of p values. It would be much better to have F values and N for the different tests. There also seems to be a contrediction between the table (effect of temperature on mass remaining), the figures (not really) and the texte (there was none).**

Table 2 has been revised to include F values and degrees of freedom for each test.

The effect of temperature on mass remaining is significant, as indicated by table 2. The text in the results section (page 6 line 28 – page 7 line 2) also indicates that mass loss and Q10 were significantly higher in the 18°C incubations. Statistics and a reference to Table 2 have been added to these statements for clarity.

Discussion of a small temperature effect is based on low Q10 compared to vascular plant decomposition (page 9, lines 6-16), despite a significant difference in mass loss between temperature treatments.

**Also the figures were not always clear as information was lacking from the legends. I do wonder if all of the figures are required, perhaps Fig 6 could be an annexe?**

We prefer to keep Fig. 6 in the main text because it clearly illustrates the lack of change in bulk C composition during incubation, which is an important conclusion but not demonstrated in the other figures. A legend has been added to Figure 6 and Figure 8 for clarity.

**In conclusion I think this is an interesting paper with a lot of potential. With a little refinement I think it could have a lot of impact.**

[revised manuscript text omitted]